# Replacement of Maize Silage and Soyabean Meal with Mulberry Silage in the Diet of Hu Lambs on Growth, Gastrointestinal Tissue Morphology, Rumen Fermentation Parameters and Microbial Diversity

**DOI:** 10.3390/ani12111406

**Published:** 2022-05-30

**Authors:** Haoqi Han, Liyang Zhang, Yuan Shang, Mingyan Wang, Clive J. C. Phillips, Yao Wang, Chuanyou Su, Hongxia Lian, Tong Fu, Tengyun Gao

**Affiliations:** 1Henan International Joint Laboratory of Nutrition Regulation and Ecological Raising of Domestic Animal, College of Animal Science and Technology, Henan Agricultural University, Zhengzhou 450046, China; hanhaoqi2021@126.com (H.H.); zhangliyang@henau.edu.cn (L.Z.); shangyuan189377@126.com (Y.S.); wmy15138236872@126.com (M.W.); wy12301112@126.com (Y.W.); suchuanyou2010@126.com (C.S.); dairycow@163.com (T.G.); 2Institute of Veterinary Medicine and Animal Sciences, Estonian University of Life Sciences, Kreutzwaldi 1, 51006 Tartu, Estonia; clive.phillips@curtin.edu.au; 3Curtin University Sustainable Policy (CUSP) Institute, Curtin University, Bentley 6102, Australia

**Keywords:** maize, mulberry, novel feeds, roughage, resource utilization, environment, Hu lambs

## Abstract

**Simple Summary:**

A shortage of high-quality roughage jeopardises the Chinese mutton sheep industry. The development of new roughage resources is important to safeguard the health and welfare of the sheep, to save costs, increase efficiency and improve resource utilization. Mulberry leaves have high nutritional value and have been used in herbivore production for a long time in China. However, fresh mulberry leaves are not easy to preserve, and dried mulberry leaves readily lose nutrients in the conservation process. Ensiling mulberry leaves can reduce the anti-nutritional constituents, mainly phytic acid and tannin, while reducing any nutrient loss. In this study, mulberry silage replaced part of a maize silage-based diet for fattening Hu lambs. The effects of mulberry silage on the growth of the lambs, their gastrointestinal tissue morphology, rumen fermentation parameters and bacterial diversity were investigated. The results showed that using mulberry silage in place of 20–40% of the maize silage in the diet of Hu lambs promoted their growth, while maintaining satisfactory digestion.

**Abstract:**

Maize silage has a significant environmental impact on livestock due to its high requirement for fertilizer and water. Mulberry has the potential to replace much of the large amount of maize silage grown in China, but its feeding value in the conserved form needs to be evaluated. We fed Hu lambs diets with 20–60% of the maize silage replaced by mulberry silage, adjusting the soybean meal content when increasing the mulberry silage inclusion rate in an attempt to balance the crude protein content of the diets. Mulberry silage had higher crude protein and lower acidic and neutral detergent fiber contents compared to maize silage. Replacing maize silage and soyabean meal with mulberry silage had no effect on the feed intake and growth rate of Hu lambs. However, the rumen pH increased, the acetate to propionate in rumen fluid increased, and the rumen ammonia concentration decreased as mulberry replaced maize silage and soyabean meal. This was associated with an increase in norank_f__F082 bacteria in the rumen. Rumen papillae were shorter when mulberry silage replaced maize silage, which may reflect the reduced neutral detergent fiber (NDF) content of the original silage. In conclusion, mulberry silage can successfully replace maize silage and soyabeans in the diet of Hu lambs without loss of production potential, which could have significant environmental benefits.

## 1. Introduction

The shortage of high-quality roughage is a limitation for the effective management of Chinese ruminant livestock industries. Conserved forages are required to feed the livestock when pasture grazing is not available. The growth of livestock and human populations in China means that there is limited land available for the production of conventional forages, such as maize silage. Ensiling a crop reduces any loss of nutrients, compared with making hay, and it is widely used in animal husbandry. Alternative sources of roughages offer the potential to improve resource utilization and reduce environmental pollution. The mulberry tree, a native of Korea and China, is widely distributed in China due to its strong drought and temperature resistance [1]. It is deep-rooted and when harvested as an animal feed has a high dry matter (DM) production in a wide range of conditions, up to 20–30 tonnes/ha/year when cut at 9–10 week intervals [2,3]. Mulberry trees are also recognized internationally as an important bioenergy and biofuel crop. They do not require irrigation, except sometimes during establishment in dry conditions.

Although it is estimated to be grown on 1.67 million ha in China, maize silage has several limitations for use as a forage for ruminants, with the short growing season a significant limiting factor in the north of the country [4]. Although yields of 20–25 t DM/ha/year are achievable [5], irrigation is often required. Maize grown for feeding to livestock has a high nitrogen demand [4], as about 300 kg/ha are harvested and 80% of this has to be applied externally [6], either in the form of artificial fertilizer or manure nitrogen. This results in large emissions of ammonia, which is responsible for major adverse effects on atmospheric pollution and human health. In the US, maize accounts for approximately 18% of total ammonia emissions, which contribute to global climate change and increase PM2.5 (particulate matter with a diameter of less than or equal to 2.5 microns in the air) to adversely affect human respiratory health. The significant fertilizer nitrogen requirements of maize also contribute to carbon dioxide emissions from fertilizer manufacture, and they are therefore important in global climate change.

Mulberry trees, by contrast, are important carbon dioxide sequestrators and relatively unresponsive to supplementary N, P and K, although good yield responses to vermicompost have been reported [7]. By developing multipurpose cultivated mulberry varieties, its utilization has been extended over 5000 years of its cultivation to fruits for Chinese medicine, cosmetics in the form of skin and hair tonics, and leaves for sericulture and livestock. Leftover branches and leaves after silkworm feeding are also sometimes fed to livestock. The nutritional value of mulberry leaves is high. Its crude protein (CP) content can be as high as 34%, with balanced amino acid composition. Its fiber content is low, and it is rich in flavonoids, sterols, polysaccharides, alkaloids and other biologically active substances [8,9,10]. Therefore it is a high-quality protein feed resource with broad application prospects [11]. Mulberry leaves are more digestible by goats [12] and sheep [13] than alfalfa hay and oat hay by sheep [14]. However, fresh mulberry leaves have a low DM content and are not easy to preserve. Dried mulberry leaves lose many nutrients, limiting their application in animal production [15]. Mulberry silage retains more nutrients while reducing the content of anti-nutritional factors in mulberry [16]. Mulberry silage may therefore have great application prospects in livestock production; however, its high content of water-soluble carbohydrates promotes clostridial activity to generate butyric acid and silage with low palatability. Therefore, ensiling should include the application of inoculants, such as cellulase and Lactobacillus casei LC [17]. 

There are no reports on the effects of mulberry silage on the rumen microflora of sheep, which is now possible to investigate with the use of high-throughput sequencing technology. We hypothesized that when used as a livestock feed, mulberry silage may have a positive effect on rumen fermentation and nutrient digestion and absorption due to its high content of polyphenols and flavonoids [18,19]. To investigate this further, we offered mulberry silage as a partial replacement for maize silage and soyabean meal in the diet of Hu lambs. In particular, we explored the effects of replacing maize silage in diets with mulberry silage on growth performance, gastrointestinal health, rumen fermentation parameters, and microbial diversity in Hu lambs, a sheep breed that originates from Mongolian sheep and is well adapted to the hot, humid conditions that prevail in central China.

## 2. Materials and Methods

### 2.1. Experimental Design

This study was conducted in Yuemeihe Agriculture and Animal Husbandry Co, Ltd., Lin Ying County, Luohe, Henan Province, China, from April to June 2021. The feeding trial was performed in accordance with the protocols approved by the Institutional Animal Care and Use Committee (IACUC) of Henan Agriculture University (Permit Number: 12-1328; Date: 05-2021). Sheep were housed in a semi-closed portal-framed building, with a transparent roof allowing good lighting of the sheep’s environment. The temperature was between 18 and 28 °C. A total of 96 healthy 3-month-old fattening Hu lambs with similar body weight (27.59 ± 3.03 kg) and physiological state were selected and divided into 4 treatment groups (6 replicates in each group, 4 animals in each replicate) according to a completely randomized design. Mulberry silage was used to replace 0% (CON group), 20% (L group), 40% (M group) and 60% (H group) of the maize silage in the Hu lambs’ diets, which were fed as a total mixed ration (TMR). To ensure that both diets were formulated to meet a 120 g/d growth rate for fattening sheep (NRC, 2007), the input of soybean meal was progressively decreased as mulberry silage replaced maize silage (Table 1) because of the higher crude protein (CP) concentration in mulberry silage. Thus, in the treatment CON, 12% (L), 24% (M) and 35% (H) of the soybean allocation was replaced by the mulberry silage. Despite this, there was an increase in ether extract, CP and Ca at high mulberry silage concentrations and a reduction in NDF and acid detergent fiber (ADF). In designing the feed composition, we aimed for a crude protein content of 14.5% (DM), but the crude protein and other nutrients in the feed changed somewhat in the different feed batches.

The feeding trial was 75 days, including a 15-day acclimation period and a 60-day formal trial, this being the normal length of the fattening period. The lambs were provided with ad libitum TMR feed twice a day at 07:00 a.m. and 17:00 p.m. Feed residues were weighed daily and the ration supply was adjusted so that 5% of the feed remained as orts each day. Each group of lambs had a separate drinking bowl providing for ad libitum intake. The body weight of each lamb was recorded (Guandong Senssun Weighing Apparatus, Zhongshan, China, error ± 0.01 kg) for two consecutive days before the morning feeding every month, and the average daily weight gain was calculated. Before the end of the experiment, 12 animals in each group were randomly selected for blood sampling, of which 6 were used for slaughter. These 6 lambs were fasted for 24 h and not given access to water for 2 h before transportation for approximately 30 min to a local slaughterhouse, complying with Regulation (EC) No. 1099/2009, where they were killed after stunning with high voltage electricity.

### 2.2. Feeding and Management of Experimental Animals

Before the experiment, the pens were sterilized, and the flocks were dewormed and sheared. Pens were disinfected with 3% Lysol twice a week. The health of the flock was checked daily, and any animals with suspected illness were treated promptly. No animals died during the test period.

### 2.3. Preparation of Mulberry Silage

Mulberry was obtained from Yuemeihe Agriculture and Animal Husbandry Development Co., Ltd, Xuchang, China. When the mulberry had grown to 1.5 m, we used a silage harvester (Claas jaguar 800, Wister, Germany) to collect the whole mulberry trees to a stubble height of 15–20 cm. The trees were harvested 3–4 times a year, with regrowth apparent 3–7 days after harvest. One gram of Chr Hansen lactococcus lactis and lactobacillus brucei powder was added to each ton of mulberry silage (lactic acid bacteria >1.3 × 10^11^ CFU/g). An automatic silage wrapping machine (Qufuxinlian Heavy Industry XL-5552, Shandong, China) was used to exclude air from the harvested material, and the samples were stored for 60 days after harvesting. After fermentation, dry matter (DM) (Association of Official Analytical Chemists AOAC, 1990: Method 934.01), ether extract (EE) (AOAC, 1990: Method 920.39), ash (AOAC, 1990: Method 942.05), calcium (Ca) (AOAC, 1990: Method 985.35), phosphorus (P) (AOAC, 1990: Method 986.24) and CP contents were determined using a Kjeldahl analyzer (Kjeltec 2300; FOSS Analytical AB, Hoganas, Sweden). Neutral and acidic detergent fibers were determined using an Ankom fiber analyzer (Ankom Technology, Fairport, NY, USA) as described by Van Soest [20]. The nutrient composition of the mulberry and maize silages is shown in Table 2.

### 2.4. Collection and Analysis of Samples

#### 2.4.1. Feed and Rumen Sample Collection and Chemical Analysis

Samples of the total mixed rations (TMR) fed for each group were collected, and their DM, EE, NDF, ADF, Ash, CP, Ca, P and gross energy (GE) were determined by the methods described above. On the 75th day of the experiment, 6 lambs in each group were randomly selected, and approximately 50 mL rumen content samples were taken by a gastric tube rumen fluid sampler (Shanghai Silidi Scientific Instrument Co., LTD., Shanghai, China) 3 h after feeding in the morning. This method has been used in previous experiments (Xue [21] and Sun [22]), which proved that the sampler and intubation method had no effect on animal health and test results. The rumen fluid samples were filtered by 4 layers of gauze and divided into 2 centrifuge tubes with 5 mL (stored in liquid nitrogen and sent to Shanghai Meiji Biotechnology Co., Ltd. (Shanghai, China) for 16S rDNA sequencing for bacterial community structure analysis). A 10 mL sample in the centrifuge tube was stored at −20 °C for NH_3_-N and volatile fatty acids (VFA) analysis. The pH of the remaining rumen fluid was immediately determined using a pH S-3C precision acidity meter (Shanghai Lei Ci Instrument Factory, Shanghai, China). The concentrations of VFA were detected using an ion chromatography system (S-150; Sykam, Munich, Germany) equipped with an chromatographic column (NaOH EGCS, Thermo Finsher, Waltham, MA, USA). The NH_3_-N concentration was analysed using the phenol hypochlorite colorimetric method [23]. Immediately after slaughter, 2 cm lengths of the jejunum and ileum and 2 cm^2^ of the rumen compartments were excised, washed with phosphate buffer and immediately fixed with 4% formaldehyde solution for observation of tissue morphology.

#### 2.4.2. qPCR Amplification of 16S rDNA Genes

All collected rumen fluid samples were subjected to high-throughput sequencing of the V3 + V4 region of the 16SrDNA gene to analyze the microbial diversity. The sequencing was performed at Shanghai Meiji Biotechnology Co, Ltd. (Shanghai, China) DNA was extracted from the samples using the Fast DNA Soil Rotation Kit (MPBio, Santa Ana, CA, USA) and high-throughput sequencing analysis was performed on the Majorbio cloud platform (www.majorbio.com, accessed on 18 March 2022). Forward primer 338F, 5′ACTCCTACGGGAGgCAGCAGCAGcag3′ and reverse primer 806R-5′-GGACTachVGGGTWTCTAAT3′ were used to amplify the 16SrDNA gene in V3 and V4 regions. Polymerase linked reaction was carried out using the methods of Wang [24]. A DNA library was constructed and run on the I11nmina MISeq instrument, and an amplification library was completed on the I11nmina MISeq PE300 platform. The NCBI Sequence Read Archive (SRA) database was used to deposit raw reads (Accession Number: PRJNA819287).

#### 2.4.3. Analysis of Gastrointestinal Tissue Morphology

Morphological examination of the intestinal tract and rumen was based on Nosworthy [25] hematoxylin-eosin staining. The tissues fixed in formaldehyde were dehydrated in ethanol (50%, 70%, 80%, 90%, 100%) solution, rinsed with xylene, and embedded in paraffin. Four 5 µm slices were cut from each well-embedded sample and stained with hematoxylin-eosin. The height of the intestinal villi, the depth of the crypt, and the length and width of the rumen papilla were measured under the microscope at 10 points each (Olympus Corporation CKX53, Tokyo, Japan). 

### 2.5. Statistical Analysis

All data were initially processed using Excel software and then SPSS software (v. 26, IBM, Armonk, NY, USA). The results of the data were analysed using the ANOVA procedure. Statistically significant differences between pairs of groups were assessed using Duncan’s test. Differences were considered significant at *p* < 0.05.

## 3. Results

### 3.1. Effects of Replacing Maize Silage and Soyabeans with Mulberry Silage on Lamb Intake and Growth

There was no significant difference in initial or final body weights or average daily gain (Table 3). There was a trend for a small increase in dry matter intake with increasing mulberry content of the diet (*p* = 0.098).

### 3.2. Rumen Fermentation Parameters

Rumen pH was higher in the group H than in CON or L (Table 4). The concentration of total volatile fatty acids (TVFA) and acetic acid was higher in H than in the other groups. Propionic acid was greater in CON than the other groups. Butyric acid was greatest in M and H and least in the CON group. TVFA production was not affected by treatment. The acetate to propionate ratio was highest for lambs in group H, intermediate in M and L, and least in the CON group. Ammonia was least in M and H, highest in CON and intermediate in L. 

### 3.3. Effects of Replacing Maize Silage and Soyabeans with Mulberry Silage on the Gastrointestinal Tissue Morphology of Lambs

The rumen papillae in the CON group were significantly taller than those of the L, M and H groups (*p* < 0.05) (Table 5, Figure 1). There were no significant treatment effects on villus height, villus width or crypt depth in the duodenum (*p* > 0.05). However, compared with the CON and H lambs, in the jejenum, the villi of lambs in L and M groups were shorter, and the crypts were shallower in the M group than the other three groups. 

### 3.4. Effects on Rumen Microbial Community Diversity

#### 3.4.1. Rumen Microbial Alpha Diversity

Based on the principle of similarity > 97%, the effective sequences in the rumen fluid were clustered and analysed. A total of 1981 OTUs were obtained. These belonged to 21 phyla, 42 classes, 95 orders, 166 families, 327 genera and 627 species. Alpha diversity analysis of the rumen microbiota showed that there were no significant differences in the Sobs index, Shannon index, Simpson index, Ace index or Chao index between the treatment groups (*p* > 0.05) (Table 6). The coverage of each group was approximately equal to 1, indicating that the sequence results covered the sample diversity. The number and composition of the rumen fluid flora of the different treatments were also similar.

#### 3.4.2. Effects on Rumen Bacterial Composition

Among the 21 phyla found in this study, the rumen-dominant phyla were Firmicutes, Bacteroidetes and Patescibacteria, accounting for approximately 98% of the total bacteria (Figure 2). The main bacterial abundances at the phylum level in groups CON, L, M, and H were Firmicutes (64, 46, 63, 52%), Bacteroidetes (28, 50, 29, 43%), and Patescibacteria (2.3, 1.4%, 2.1%, 1.5%). The bacterial abundances of Firmicutes, Bacteroidetes and Patescibacteria were similar in treatments CON and M, but higher than in treatments L and H.

A total of 327 taxa were found in the analysis at the genera level of the rumen flora. The relative abundance of rumen bacteria in each group (abundance > 1%) is shown in Figure 3A. Prevotella, Ruminococcus, Christensenellaceae-R-7-group and the Rikenellaceae-RC9-gut-group, etc., were the dominant bacteria in the rumen fluid of each group, accounting for approximately 90% of the total bacteria. The abundances of Prevotella in the CON, L, M and H groups were 18.94, 33.42, 18.88 and 36.34%, Ruminococcus 19.33, 5.49, 18.77 and 20.47%, Christensenellaceae-R-7-group 6.87, 16.12, 5.61, 5.58%, and Rikenellaceae-RC9-gut-group 2.77, 9.51, 4.50, and 3.13%, respectively. The abundances of norank_f__F082, Papillibacter, UCG-001, and UCG-009 in the CON group were significantly higher than in the L, M, and H groups (*p* < 0.05). The abundances of norank_f__Prevotellaceae, Prevotellaceae_UCG-001, Quinella and unclassified_p__Firmicutes were significantly higher in the M group than those in the CON, L, and H groups (*p* < 0.05). The abundance of norank_f__Bifidobacteriaceae and Flexilinea in the L group was significantly higher than those of the CON, M and H groups (*p* < 0.05). The differences in the rumen genera levels of lambs in each group are shown in Figure 3B.

#### 3.4.3. Multilevel Species Difference Discrimination Analysis

To determine the functional communities in the samples, LEfSe technology was used to analyze the specific microorganisms in the rumen fluid of the four groups of lambs (Figure 4A,B). Microorganisms with an LDA score > 2 were assumed to be specific microorganisms among the microorganisms that differentiated the treatment group from the other treatment groups. Thirty-three specific microbial genera were found, of which ten genera were significantly enriched in the CON group, twelve genera were enriched in the L group, and eleven genera were enriched in the M group (Figure 4). These genera had a significant impact on sample grouping.

#### 3.4.4. Relationship between Environmental Variables and Microbial Diversity and Abundance

Figure 5 shows the correlations between the relative abundance of genera level rumen bacteria (top 20 genera) and rumen fermentation parameters. There was a significant negative correlation between pH and the relative abundance of NK4N214_group (R = −0.547, 0.001 < *p* < 0.01). The relative abundance of norank_f__F082 was positively correlated with propionic acid content (R = 0.541, *p* < 0.05) and NH_3_-N content (R = 0.636, *p* < 0.001). PH (R = −0.460, *p* < 0.05) and butyric acid (R = 0.487, 0.001 < *p* < 0.05) were negatively correlated with norank_f__F082.

## 4. Discussion

### 4.1. Effects of Mulberry Silage on the Growth Performance of Lambs

Mulberry silage had good palatability. In this study, the dry matter intake and average daily gain (ADG) of lambs in the L and M groups were as good as those in the CON group, which was similar to the results in studies conducted by Sheng [26], Hua [27] and Tao [28], who used dried mulberry leaves and mulberry silage to feed sheep and beef cattle. Using mulberry silage to replace up to 60% of maize silage in the diet is an effective way to introduce a more environmentally friendly crop. Maize has a significantly greater requirement for fertilizer than mulberry, a similar cost of harvest but a lower yield than mulberry. Due to the contribution of maize to environmental pollution, measures are needed to replace maize with other crops with similar yield potential. Using mulberry there are likely to be less CO_2_ and ammonia emissions, which may be more conducive to reducing environmental pollution and the sustainable utilization of resources.

### 4.2. Effects of Mulberry Silage on Rumen Fermentation Parameters of Lambs

The normal range of rumen pH is 6.0–7.5, and the changes in the pH are related to ruminant salivary secretion, chyme outflow velocity and composition [29,30,31], which can be used as a comprehensive index to evaluate the level of rumen fermentation. In this study, the pH gradually increased with the increase in the proportion of mulberry silage, but it was all within the normal range. The increase likely reflects VFA changes. VFAs provide energy for the maintenance and growth of ruminants. Any changes directly affect the digestion, absorption and utilization of nutrients by ruminants [32], and their generation is affected by diet composition, processing mode, rumen microbes and feeding mode [33]. With the increase in mulberry silage proportion, propionic acid content gradually decreased, butyric acid content gradually increased, and acetic acid increased with the highest level of mulberry inclusion. This is broadly consistent with the research results from Wang [34], who fed mulberry silage to goats. Mulberry silage therefore changes the rumen fermentation pattern in lambs to increase the acetate to propionate ratio. The changes in rumen VFA across treatments appears to have been mainly influenced by the Bacteroidales norank_f__F082, which was strongly positively correlated with ammonia and propionic acid. Ammonia, propionic acid and norank_f__F082 were reduced in the mulberry substitution treatments. It was also negatively correlated with butyric acid and pH, both of which were elevated in the mulberry substituted treatments. It has been found to be associated with high butyric acid concentration in calves [35].

Treatment had no effect on the concentration of Ruminococcus, even though Ruminococcus albus and Ruminococcus flavanum were associated with the digestion of cellulase and hemicellulose. There were differences between treatments in NDF and ADF concentrations in the diet. Acetic acid is the main product of fibrolytic bacteria, and there is a positive correlation between acetic acid concentration and the abundance of Ruminococcus. Ruminal NH_3_-N reflects the degradation of proteins, peptides and amino acids in the diet [36]. In this study, the NH_3_-N concentration decreased with the increase in mulberry inclusion rate, which was consistent with the results obtained by Ouyang [37] and Wang [34]. The decrease in NH_3_-N concentration may be due to the increase in rumen pH after the feeding of mulberry silage, which promoted the abundance and activity of ammonium-based microbes, consistent with the greater abundance of Prevotella in the L and H groups. The experiment suggests that adding mulberry silage to the diet could improve the utilization efficiency of nitrogen or that the degradation of protein in the rumen was limited.

### 4.3. Effects of Mulberry Silage on Gastrointestinal Tissue Morphology of Lambs

The villus height and crypt depth of the small intestine are closely related to the digestion and absorption function of the intestine. The height of the small intestine villi represents the amount of contact between the intestinal tract and the chyme. The more contact with the chyme, the more adequate the absorption of nutrients. The deeper the crypts are, the greater the atrophy of the villi and the less the intestinal absorption capacity [38]. Villus height/crypt depth (V/C) can reflect the functional state of the small intestine. The higher the V/C is, the stronger the intestinal absorption capacity [39]. In this study, there was no significant difference in duodenal villus height, villus width, crypt depth or V/C between the experimental groups, indicating that mulberry silage had no adverse effect on duodenal development in fattening lambs. The growth and development of the rumen tissue in ruminants experience corresponding morphological changes according to the diet’s physical shape, type and nutritional level to meet the needs of the normal physiological metabolism of the body [40,41]. In this study, the length and width of ruminal papilla decreased gradually with the addition of mulberry silage. This may be because NDF in the diet decreased with the addition of mulberry silage, which reduced the contact between the NDF in feed and the rumen papilla. This may delay the growth and development of the rumen [31].

### 4.4. Effects of Mulberry Silage on the Rumen Microbial Composition of Lambs

This study used the Majorbio sequencing platform to investigate the effect of mulberry silage replacing maize silage on the rumen microflora. The Ace and Chao indices represent the relative abundance of microbial communities; the larger the Shannon index value, the smaller the Simpson index value, indicating a higher species diversity in the sample. However, the inclusion of mulberry silage did not cause significant changes in the Ace, Chao, Shannon or Simpson indices, indicating that mulberry silage had no effect on microbial diversity.

The complex rumen microbiota plays an important role in the host’s absorption of nutrients in the feed [42], and its composition is influenced by dietary composition and processing methods, etc. [43]. This study categorized comprehensively the structure of the rumen microbial community of fattening lambs. Firmicutes and Bacteroidetes, involved in carbohydrate and protein degradation, were the two most abundant phyla [44], which is consistent with the results of the study by Derakhshani [45]. The effects of mulberry silage on bacterial populations at the genera level showed that the abundance of Prevotella in the L and H groups was higher than that of the CON group. Prevotella species can use a variety of substrates (such as starch, proteins and hemicellulose) to produce succinate and formate [46]. The substantial metabolic diversity of this genera is indicated by the presence of Prevotella spp. in the rumen across the different diets [47]. The results showed that the addition of mulberry silage to the diet affected the composition and abundance of Prevotella, suggesting improvements in the decomposition efficiency of nonfiber polysaccharides, proteins and other nutrients. This was significantly correlated with the growth performance, NH3-N concentration, norank_f__Prevotellaceae and Prevotellaceae_UCG-001 in the rumen of fattening lambs mentioned above. This study also found significant differences in the low abundance of UCG-001, UCG009, Quinella and norank_f__F082 in each treatment, which is likely to have a specific important function. The function of this bacterium in determining whether it is related to the absorption and utilization of mulberry silage in the rumen of fattening lambs will be further investigated.

### 4.5. Relationship between Rumen Microorganisms and Fermentation Parameters of Lambs

Nutrient utilization in feed mainly depends on the rumen microbiota [43], and differences in rumen fermentation parameters may be indicators of feed utilization [48]. One aim of this study was to investigate the relationship between the rumen microbiota and rumen fermentation parameters. The abundance of NK4A214_group was negatively correlated with pH, indicating that the increase in rumen pH after feeding with mulberry silage affected the number and activity of NK4A214_group. Since norank_f__F082 was associated with rumen fermentation parameters, we speculate that the norank_f__F082 genera flora played an important role in the digesting of mulberry silage in the rumen. Notably, a large number of ruminal bacteria in the tested lambs belonged to unclassified and non-ranking genera. This suggests that lambs fed mulberry silage might have a more abundant ruminal microbe, and only a small portion of the bacteria have been identified with next-generation sequencing technology.

## 5. Conclusions

Compared with maize silage, mulberry silage had some nutritional advantages, even though these were not reflected in body weight gain. The replacement of maize silage and soyabean meal with mulberry silage improved the rumen fermentation pattern, and possibly increased the utilization rate of nitrogen. Its reduced fiber content appears to have retarded rumen papillae development, which may explain why the improved nutrient content did not improve weight gain. Although mulberry silage can cause changes in the rumen histological morphology and the rumen microbial composition, the bacterial diversity indices did not change significantly, indicating that the rumen fermentation status remained stable after adding mulberry silage to the diets. Therefore, replacing maize silage with mulberry silage offers great potential in reducing the environmental impact of livestock production without sacrificing production capacity.

## Figures and Tables

**Figure 1 animals-12-01406-f001:**
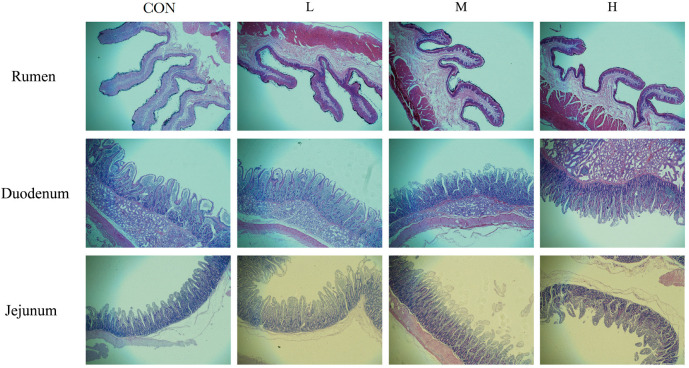
Histological observation the lambs’ rumen, duodenum and jejunum tissues. CON, L, M and H represent the substitution of 0, 20, 40 and 60% of maize silage with mulberry silage, respectively.

**Figure 2 animals-12-01406-f002:**
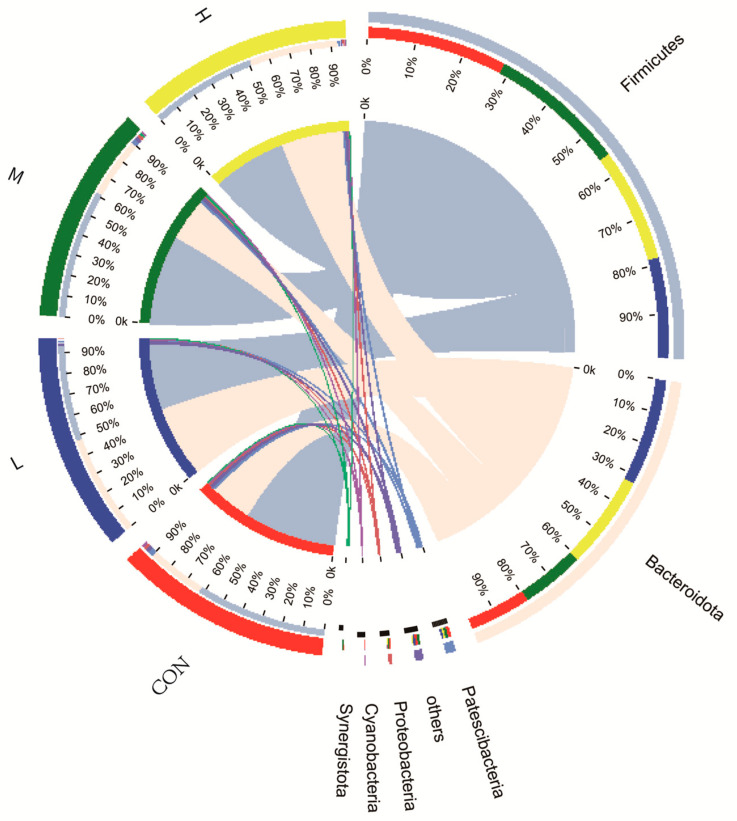
Composition of rumen microbial phyla in lambs. CON, L, M and H represent the substitution of 0, 20, 40 and 60% of maize silage with mulberry silage, respectively.

**Figure 3 animals-12-01406-f003:**
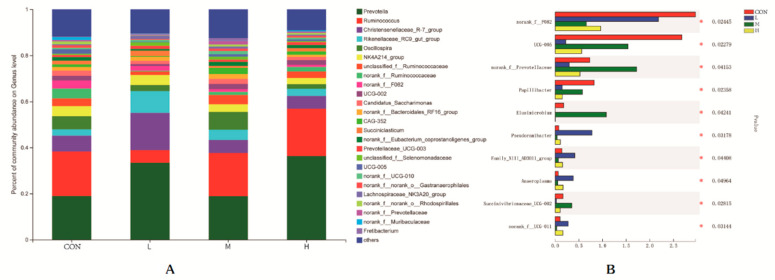
Composition (**A**) and difference (**B**) of rumen microbial genera level in lambs. Only the dominant genera (those with relative abundances ≥ 1%) of rumen bacteria are shown. Corrected *p* < 0.05 is marked with *. CON, L, M and H represent the substitution of 0, 20, 40 and 60% of maize silage with mulberry silage, respectively.

**Figure 4 animals-12-01406-f004:**
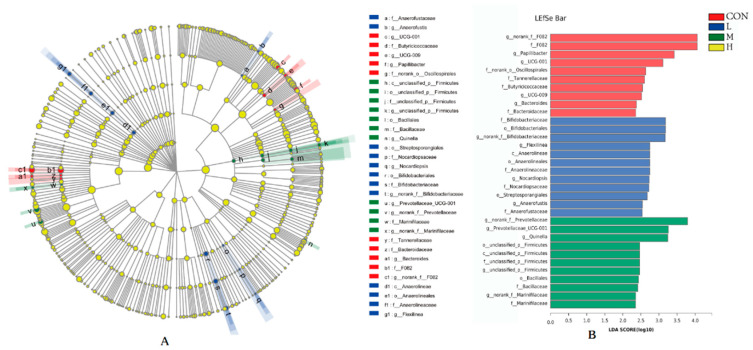
Bacteria differentiated between samples, different groups identified by linear discriminant analysis (**A**). Green, red and blue bars or nodes represent the bacteria with the significantly higher relative abundance in CON, L and M groups, respectively. LDA analysis of rumen microbial abundance (**B**). The larger the LDA score, the greater the influence of species abundance on the difference effect.

**Figure 5 animals-12-01406-f005:**
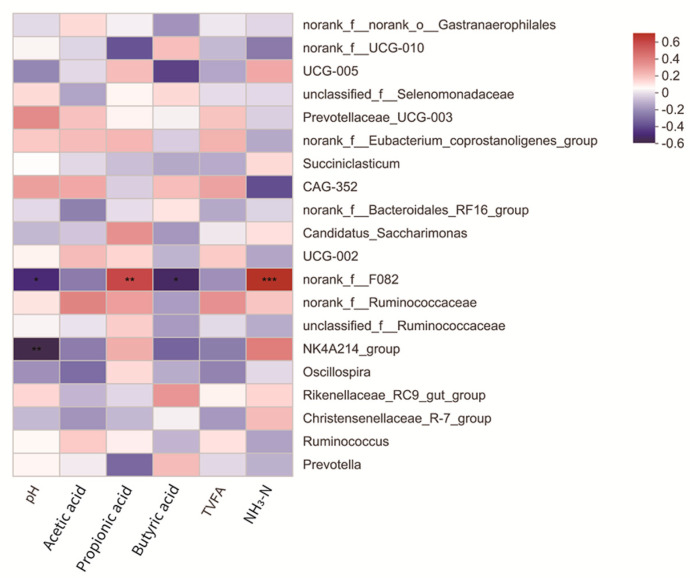
Relationship between environmental variables and microbial diversity and abundance. R values are displayed in different colors, *, ** and *** represent *p* < 0.05, 0.01 and 0.001, respectively, and the legend on the right is the colour interval of different R values.

**Table 1 animals-12-01406-t001:** Composition and nutrient levels of basal diets.

Items	Treatment ^3^			
	CON	L	M	H
Ingredients (g/kg DM)				
Maize	200	220	240	260
Wheat bran	100	100	100	100
Soybean meal	170	150	130	110
Maize silage	250	200	150	100
Mulberry silage	0	50	100	150
Peanut vines	250	250	250	250
Premix ^1^	20	20	20	20
Bicarbonate of soda	10	10	10	10
Total	1000	1000	1000	1000
Nutrient concentrations ^2^ (g/kg DM, except where otherwise stated)				
Dry matter (g/kg fresh weight)	445.1	444.1	443.2	441.8
Digestible Energy (DE), MJ/kg	12.3	12.0	12.0	11.8
Crude protein (CP)	135.6	137.1	142.8	149.3
Ether extract (EE)	27.7	29.5	28.9	30.8
Neutral detergent fiber (NDF)	394.7	378.2	370.3	367.8
Acid detergent fiber (ADF)	250.8	241.0	238.3	225.8
Ash	122.8	116.5	117.4	109.4
Calcium (Ca)	13.3	13.2	15.2	16.5
Phosphorus (P)	2.2	2.2	2.2	2.5

^1^ The premix provided the following per kg of the concentrate: Vitamin A 1.0 × 10^4^ IU, Vitamin D3 2000 IU, Vitamin E 125 IU, Niacin 250 mg, pantothenic acid 75 mg, Biotin b 5mg, Cu 20 mg, Fe 68 mg, Mn 56 mg, Zn 50 mg, I 1.05 mg, Se 0.2 mg, Co 0.75 mg. ^2^ Nutrient levels were measured values. ^3^ CON, L, M and H represent the substitution of 0, 20, 40 and 60% of maize silage with mulberry silage, respectively.

**Table 2 animals-12-01406-t002:** Nutrient composition of mulberry and maize silages (g/kg DM, except where otherwise stated).

Items	DM (g/kg Fresh Weight)	EE	CP	Ash	NDF	ADF	Ca	P	GE (MJ/kg)
Mulberry silage	247.6	73.8	169.0	87.2	429.9	284.2	11.7	4.0	17.05
Maize silage	308.1	45.4	78.0	80.1	500.7	298.7	6.2	1.1	16.51

DM dry matter; EE ether extract; CP crude protein; NDF neutral detergent fibers; ADF acidic detergent fibers; Ca calcium; P phosphorus; GE gross energy.

**Table 3 animals-12-01406-t003:** Effects of replacing 0 (CON group), 20 (L group), 40 (M group) and 60% (H group) of maize silage with mulberry silage on lambs’ feed intake and growth.

Items	Treatment				SEM ^1^	*p*-Value
	CON	L	M	H		
Initial body weight, kg	27.38	27.94	27.05	28.01	0.36	0.752
Final body weight, kg	41.21	42.28	41.45	41.67	0.41	0.828
Total weight, kg	13.83	14.34	14.40	13.66	0.21	0.528
Dry matter intake, g/d	1168	1170	1175	1178	0.08	0.098
Average daily gain, g/d	231	239	240	228	3.52	0.528
Feed conversion ratio (DM intake/live weight gain)	5.13	4.98	5.01	5.20	0.07	0.447

^1^ SEM (Standard error of mean).

**Table 4 animals-12-01406-t004:** Effects of replacing 0 (CON group), 20 (L group), 40 (M group) and 60% (H group) of maize silage with mulberry silage on lambs’ rumen fermentation characteristics.

Items	Treatment				SEM ^2^	*p*-Value
	CON	L	M	H		
pH	6.38 ^b^	6.38 ^b^	6.47 ^ab^	6.57 ^a^	0.22	0.004
Acetic acid, mmol/L	87.74 ^b^	85.58 ^b^	85.25 ^b^	93.29 ^a^	1.28	0.021
Propionic acid, mmol/L	35.51 ^a^	31.58 ^b^	29.56 ^b^	28.13 ^b^	0.91	0.001
Butyric acid, mmol/L	13.55 ^c^	19.11 ^b^	22.27 ^a^	24.53 ^a^	0.88	<0.001
TVFA ^1^, mmol/L	136.80	136.54	137.08	145.95	1.78	0.366
Acetate: Propionate ratio	2.47 ^c^	2.71 ^bc^	2.89 ^b^	3.31 ^a^	0.05	<0.001
NH_3_-N, mg/dL	76.40 ^a^	56.70 ^b^	21.91 ^c^	21.49 ^c^	4.42	<0.001

^1^ TVFA (total volatile fatty acids) = acetate + propionate + butyrate. ^2^ SEM (Standard error of mean). Within rows, mean values with different superscripts differ significantly by Duncan’s Multiple Comparison Test (*p* < 0.05).

**Table 5 animals-12-01406-t005:** Effects of replacing 0 (CON group), 20 (L group), 40 (M group) and 60% (H group) of maize silage with mulberry silage on lambs’ gastrointestinal tissue morphology.

Items	Treatment				SEM ^1^	*p*-Value
	CON	L	M	H		
Rumen						
Papilla height, μm	1981.43 ^a^	1542.02 ^b^	1553.12 ^b^	1457.79 ^b^	68.72	0.014
Papilla width, μm	401.44	398.26	374.29	360.14	9.15	0.346
Duodenum						
Villus height, μm	1039.92	946.82	949.95	959.96	30.30	0.705
Villus width, μm	168.73	199.59	173.96	177.08	6.70	0.410
Crypt depth, μm	416.42	424.97	386.58	373.03	17.57	0.735
V/C ^2^	2.55	2.28	2.28	2.65	0.12	0.790
Jejunum						
Villus height, μm	866.63 ^a^	790.75 ^b^	709.59 ^c^	900.34 ^a^	21.26	<0.001
Villus width, μm	183.91	161.55	153.14	177.08	5.60	0.192
Crypt depth, μm	741.89 ^a^	650.01 ^a^	459.80 ^b^	744.98 ^a^	32.76	<0.001
V/C	1.17 ^b^	1.22 ^b^	1.55 ^a^	1.22 ^b^	0.05	0.001

^1^ SEM (Standard error of mean).^2^ V/C (Villus height/Crypt depth). Within rows, mean values with different superscripts differ significantly by Duncan’s Multiple Comparison Test (*p* < 0.05).

**Table 6 animals-12-01406-t006:** Effects of replacing 0 (CON group), 20 (L group), 40 (M group) and 60% (H group) of maize silage with mulberry silage on lambs’ rumen microbial alpha diversity (n = 6).

Items	Treatment				SEM ^1^	*p*-Value
	CON	L	M	H		
Sobs index	784.33	806.50	776.00	721.50	23.59	0.653
Shannon index	4.42	4.41	4.54	4.14	0.10	0.603
Simpson index	0.05	0.05	0.03	0.07	0.01	0.554
Ace index	979.02	1067.57	976.46	967.89	31.72	0.680
Chao index	985.85	1053.54	976.08	945.81	29.26	0.642
Coverage index	0.99	0.99	0.99	0.99	0.01	0.446

^1^ SEM = Standard error of mean.

## Data Availability

The data used in this study are available from the corresponding author on request.

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
