# Peer review of "Replacement of Maize Silage and Soyabean Meal with Mulberry Silage in the Diet of Hu Lambs on Growth, Gastrointestinal Tissue Morphology, Rumen Fermentation Parameters and Microbial Diversity"

_animals, 2022, doi:10.3390/ani12111406_

Round 1

Reviewer 1 Report

Replacement of maize silage with mulberry silage in the diet of Hu sheep on growth, gastrointestinal tissue morphology, rumen fermentation parameters and microbial diversity

Comments

Abstract  

  1. Why did you choose for silage preparation from mulberry leaves instead of feeding fresh or dry leaves, whether it is suited for silage preparation? Pl. give proper justification?
  2. Economics of feeding mulberry silage to sheep was not under taken in this study?

Materials and methods

L 118: What was the age of experimental animals? Whether adult sheep were used for the study, otherwise sheep should be replaced by lambs throughout the manuscript, if the animals were in growing stage?

L 128: What is treatment 0? Pl. define it properly?

L 131: The duration of the growth trial (75) is very short for this kind of study?

L 287: Authors adjusted the offered feed in such a way that animal should refuse 5% orts each day? However, for ad libitum feeding animals should refuse at-least 10-15% of the offered amount? Pl. give proper justification for this adjustment?

L 235: The treatment/group 0 may be designated as control throughout the manuscript?

Table3: Initial and final body weights, DMI and OMI g/d and g/kg metabolic size may also be provided?

discussion

L 364: The productivity of mulberry leaves as compared to maize and collection cost of leaves are important factors that should have been considered before arising any conclusion?

L 375: The increase is likely to reflect VFA production. But it was not reflected on the performance of animals?

L 403: If nitrogen utilization efficiency was improved with mulberry silage feeding, it should have been reflected on the performance of sheep?

Conclusion

Mulberry silage replacement of maize silage offers major potential to reduce the environmental impact of livestock production. However, per kg cost of production of mulberry leaves should be taken into consideration?

Besides these some corrections have been suggested on the body of the text.

Author Response

Comments

Abstract

  1. Why did you choose for silage preparation from mulberry leaves instead of feeding fresh or dry leaves, whether it is suited for silage preparation? Pl. give proper justification?

Answer: Conserving feed as silage enables farmers to feed mulberry leaves that were harvested at a single time, thereby avoiding the need for regular harvest of fresh material. Dried leaves have substantial losses of dry matter and nutrients in the conservation process, including leaf shatter and leaf loss.

  1. Economics of feeding mulberry silage to sheep was not under taken in this study?

Answer: Agreed, we have removed this.

Materials and methods

L 118: What was the age of experimental animals? Whether adult sheep were used for the study, otherwise sheep should be replaced by lambs throughout the manuscript, if the animals were in growing stage?

Answer: The Hu sheep used in the experiment are 3 months old, which has been modified in the text.

L 128: What is treatment 0? Pl. define it properly?

Answer: Group 0 is the control group, which has been modified in the text.

L 131: The duration of the growth trial (75) is very short for this kind of study?

Answer: At the end of this experiment, the weight of the sheep had reached the market standard and they could be slaughtered and sold. Thus it represents a typical growth period before slaughter.

L 287: Authors adjusted the offered feed in such a way that animal should refuse 5% orts each day? However, forad libitum feeding animals should refuse at-least 10-15% of the offered amount? Pl. give proper justification for this adjustment?

Feed surplus is unavoidable in the process of animal feeding, and undoubtedly a large amount of feed surplus is very wasteful, which increases the cost of breeding and environmental pollution. Therefore, we believe that a lower amount of leftover feed may be of great significance to improve feeding efficiency and reduce environmental pollution. The animals in other studies used the 5% feed rejection amount:

Hu Gaojie. etal. Effects of Ginkgo Leaves on Growth Slaughter Performance and Nutrient Ultilization in Sheep. Chinese animal husbandry and veterinary medicine. 2020,47(04),1041-1049

Mao Yafang,etal. Effects of a-Lipoic Acid and Yeast Chromium on Growth Performance,Plasma Biochemical Indexes and Nutrient Digestion and Utilization of Sheep under Heat Stress. Chinese Journal of Animal Nutrition,2022,32(09),4212-4221.

L 235: The treatment/group 0 may be designated as control throughout the manuscript?

Answer: The full text has been revised to indicate that this is a Control treatment.

Table3: Initial and final body weights, DMI and OMI g/d and g/kg metabolic size may also be provided?

Answer: The relevant data is now provided in the table.

discussion

L 364: The productivity of mulberry leaves as compared to maize and collection cost of leaves are important factors that should have been considered before arising any conclusion?

Answer: In the process of growing the crop, maize requires a lot of chemical fertilizers, while mulberry are less dependent on chemical fertilizers. The harvesting costs of the two are similar, but the yield of mulberry is potentially higher than that of maize, so mulberry may have higher benefits.

L 375: The increase is likely to reflect VFA production. But it was not reflected on the performance of animals?

Answer: The increase of pH may be related to the increase of TVFA content, but in this study, no significant relationship was found between the increase of TVFA and the growth performance of Hu sheep, so you are correct. We hypothesize that the quality of the diet was sufficient to allow maximum growth.

L 403: If nitrogen utilization efficiency was improved with mulberry silage feeding, it should have been reflected on the performance of sheep?

Answer: Increased nitrogen use efficiency may have improved animal performance, but the lower fiber content of mulberry silage may have inhibited the development of rumen papillae, which may explain the increased nitrogen use efficiency without improved growth performance.

Conclusion

Mulberry silage replacement of maize silage offers major potential to reduce the environmental impact of livestock production. However, per kg cost of production of mulberry leaves should be taken into consideration?

Answer: This study was focused on the nutritional comparison of mulberry and maize. A different study would be required to focus on the broader implications, such as cost of growing the crop.

Reviewer 2 Report

The aim of the research was to determine the effect of replacement of maize solage with mulberry silage in the diet of Hu Steep on growth, gastrointestinal microstructure, rumen fermentation parameters and microbial diversity. The number sheep used in the experiment is sufficient. The applied research methods are correct. The discussion is well conducted and comprehensive. Well-chosen references. Before publishing in Animals, the article requires additions and corrections. The proposed changes are listed below:

General comments:

Please prepare the article in accordance with the instructions for authors.

  • When describing the significant, use lowercase "p" in italics, spaces before and after "<" for example (p < 0.05)
  • Please use the abbreviated name journal for item number 2, 6, 12, 17, etc.
  • Abbreviated name journal must be correct
  • There is no journal name, volume number, or page numbers for several references, please correct them.
  • There must be "dot" after every parts abbrevaited name journal, for example, "Poult. Sci." Instead of Poult Sci
  • In the References section, for a range of pages, use the long "-" from the insert function for all References items

Detailed comments:

L29 20-40% instead of 20% -40%

L53 + please write something about the importance of silages in the nutrition of farm animals, why and when they are used (when there is no forage?), Which silages are most often used in China, apart from maize silage

L58 spp or spp. ?

L69 space before [4]

L91 space before [13]

L113 add information about housing animals: closed building, without windows? What kind of bedding, what temperature, humidity, length, color and intensity of light, what dimensions of the pen

L135 "Hu" instead of "hu"

L137 what kind of balance, name, accuracy of measurements BW

L143 how the sheep were slaughtered, briefly describe the method

L146-147 after the number, a space before the measurement unit

L152 what measure, what concentration, how

L157 space after 1.5

L171 delete P J

L183 delete D, and J by surname

L196 cm2?

L227, 232, 234, etc. "p" lowercase and italic

In table 3-6 "p-Value" instead of "P value"

L232 No ADG description

L 246-247 Description not in accordance with the data in Table 4

In table 5, use the top index for "Papilla height" and "Villus height" for significance

L279 „p” in small and in italic

Figure 3A and 3B have larger dimensions, one below the other

Figure 4A and 4B larger dimensions, one below the other

L357 delete "P" and "J" by surname

L358 delete "H"

L380 delete "Yaoyue"

L389 space before [35]

L398 delete "Jialiang" and "Ouyang" leave only the surnames

L476 body weight gain instead of weight gain

References position 3 no volume

L519 "ek"?

L520 missing journal name, volume number, and page range

L523, 526, etc. a comma instead of a colon before the page numbers

L530 no journal name

L535 missing volume number

L543, 577 „Asian Australas. J. Anim. Sci.” instead of current form

L571 "Sci." instead of „Sc, i”

L588 Abbreviated name journal must be

L601, 607, 620, 623, 629 a comma instead of a colon before the page range

Author Response

General comments:

Please prepare the article in accordance with the instructions for authors.

  • When describing the significant, use lowercase "p" in italics, spaces before and after "<" for example (p< 0.05)
  • Please use the abbreviated name journal for item number 2, 6, 12, 17, etc.
  • Abbreviated name journal must be correct
  • There is no journal name, volume number, or page numbers for several references, please correct them.
  • There must be "dot" after every parts abbrevaited name journal, for example, "Poult. Sci." Instead of Poult Sci
  • In the References section, for a range of pages, use the long "-" from the insert function for all References items

Answer:The full text and the incorrectly formatted part of the bibliography have been revised

Detailed comments:

L29 20-40% instead of 20% -40%

Answer:This has been corrected, We have implemented this change.

L53 + please write something about the importance of silages in the nutrition of farm animals, why and when they are used (when there is no forage?), Which silages are most often used in China, apart from maize silage

Answer: We have added that High-quality roughage is in short supply and expensive in China, and the use of some unconventional forages without affecting the growth of animals may reduce large farming costs. Modified in text as requested.

L58 spp or spp. ?

Answer:We have implemented this change

L69 space before [4]

Answer:We have implemented this change.

L91 space before [13]

Answer:We have implemented this change.

L113 add information about housing animals: closed building, without windows? What kind of bedding, what temperature, humidity, length, color and intensity of light, what dimensions of the pen.

Answer:The house was a semi-closed double-slope building, the temperature was maintained at 18-28 degrees, the north and south faces were transparent, and the lighting was good.

L135 "Hu" instead of "hu"

Answer:We have implemented this change.

L137 what kind of balance, name, accuracy of measurements BW

Answer: We have added the required information.

L143 how the sheep were slaughtered, briefly describe the method

Answer: These sheep were fasted for 24 h and not given access to water for 2 h before transportation for approximately 30 min to a local slaughterhouse, complying with Regulation (EC) No. 1099/2009. The Hu sheep were slaughtered after stunning with high voltage electricity.

L146-147 after the number, a space before the measurement unit

Answer:We have implemented this change.

L152 what measure, what concentration, how

Answer: Relevant information has been added.

L157 space after 1.5

Answer:We have implemented this change.

L171 delete P J

Answer:We have implemented this change.

L183 delete D, and J by surname

Answer:We have implemented this change.

L196 cm2?

Answer:Correct, square centimeters.

L227, 232, 234, etc. "p" lowercase and italic

In table 3-6 "p-Value" instead of "P value"

Answer:We have implemented this change.

L232 No ADG description

Answer:Since ADG did not differ significantly between groups, it was not described.

L 246-247 Description not in accordance with the data in Table 4

Answer:We have corrected this misrepresentation.

In table 5, use the top index for "Papilla height" and "Villus height" for significance

Answer:We have implemented this change.

L279 „p”in small and in italic

Answer:We have implemented this change.

Figure 3A and 3B have larger dimensions, one below the other

Figure 4A and 4B larger dimensions, one below the other.

Answer:Image has been reformatted .

L357 delete "P" and "J" by surname

Answer:We have implemented this change

L358 delete "H"

Answer:We have implemented this change.

L380 delete "Yaoyue"

Answer:We have implemented this change.

L389 space before [35]

Answer:We have implemented this change.

L398 delete "Jialiang" and "Ouyang" leave only the surnames

Answer:We have implemented this change.

L476 body weight gain instead of weight gain

Answer:We have implemented this change.

References position 3 no volume

Answer:We have implemented this change.

L519 "ek"?

Answer:We have implemented this change.

L520 missing journal name, volume number, and page range

Answer:We have implemented this change.

L523, 526, etc. a comma instead of a colon before the page numbers

Answer:We have implemented this change.

L530 no journal name

Answer:We have implemented this change.

L535 missing volume number

Answer:We have implemented this change.

L543, 577 „Asian Australas. J. Anim. Sci.” instead of current form

Answer:We have implemented this change.

L571 "Sci." instead of „Sc, i”

Answer:We have implemented this change.

L588 Abbreviated name journal must be

Answer:We have implemented this change.

L601, 607, 620, 623, 629 a comma instead of a colon before the page range

Answer:We have implemented this change.

Reviewer 3 Report

General points about the manuscript: The manuscript brings interesting information about the effect of mulberry silage replacing maize silage of sheep diet on growth characteristics, gastrointestinal tissue morphology, rumen fermentation parameters and bacterial diversity. This topic is relevant to comprehend the effects of inclusion levels of mulberry silage on different physiological and nutritional aspects of sheep. In my opinion the manuscript has a good aim and the manuscript is well written.

My biggest concern is that the composition of other ingredients in the diets also changed, such as maize and soybean (not only maize silage). So, in fact the study is not only “Replacement of maize silage with mulberry silage”. The other effects might confound the results and thus should be taken into account in the title and in the text. Additionally, the crude protein of the diets varied from 13.5% to 14.9%, this could be another confounding effect and authors must give a strong explanation on it.

Specific considerations:

Keywords: Do not use as keywords those words already mentioned in the title.

L74: Please better describe the meaning of PM2.5, I wonder that many readers are not familiar with it.

“Mulberry trees, by contrast” is repeated twice (L75 and L79), please re-write on of the sentences in order to sound linguistically better.

This manuscript is a bit of a mixture of British (BT) and American (AM) English, please be consistent and choose only one style. Maybe the one the Journal follows the most. Examples: Analysed (BT) vs. Analyzed (AM); fibre (BT) vs. fiber (AM); alfalfa (AM), if it would be BT spelling, then it would be lucerne.

L107: “effect of different mulberry silage on the ability of mulberry silage…” I believe that this requires proofreading and re-writing.

L132: Please describe more clearly how the experiment was conducted and the sampling was done along the 60-day period. For example, were samples taken in the last week of this period?

L142: Please use “time of slaughtering”.

L161: The factorial number is not correct formatted. Please put it as superscript. Additionally describe better what kind of inoculant you used. What are the strains in it? Homo or heterofermenter?

Table 2: Abbreviations in the table should be spelled out in footnote. Additionally, “freshweight” is written all together, please put a space.

Fermentation quality (pH, NH3-N, VFAs and so on) of the silages were not mentioned, however this is a very important information. Is it already published? If so, please just indicate the published paper, otherwise, please include it in this manuscript.

L193: “column” is written twice, delete one of them.

L196: Use superscript in cm2.

L209: Is this correct “I11nmina”? Please check and fix.

L215: If this abbreviation is not used again, it’s not needed to be defined.

L222: As there are 4 treatments, authors can also investigate linear, quadratic and cubic effects of mulberry replacement level on the diet of sheep.

Table 3: “liveweight” is written all together, please put a space. Abbreviations in Table were defined, but not used, so better not to define them. Should SEM be spelled out in footnote? If so, please check throughout.

L238: The description of treatments was already given in the Title.

L242: VFA was already previously described. Please describe abbreviation only first time mentioned and then use just the abbreviation onward. Please check throughout for other abbreviations as well.

L244: It’s missing an end period in a sentence.

L246: This sentence “Ammonia was highest in M and H” does not match the results in Table 4, please check and fix.

Table 4: Please put the title of this table in line with the title of Table 3. The same for the titles of Tables 5 and 6.

Table 5: Put the Duncan’s letters in superscript. Use the correct symbol for “um”. What is “V/C”? Define. Footnote is missing end period.

Mention Figure 1 in text before inserting the Figure.

Figure 3: Isn’t the H treatment (yellow) in the Figure? Why?

L343: “was associated”, please indicate the direction of the association (positive or negative).

L344: “NH3-N content was positively correlated with pH and butyric acid content”, the Figure does not allow this affirmation. Please check and fix.

L357: Please check journal’s guideline on the way of including the references in the text. It’s not consistent at the moment. Please also add space before and after [ ]. Check throughout the manuscript.

L364: Please put the number 2 in CO2 subscript.

L386: Positively? Wasn’t it negatively correlated?

L391: Scientific names should be in italic. Check throughout.

L411: Was V/C previously described?

L465: This was already mentioned in L384. Check and fix the text in one of these places, but do not repeat it. Maybe the best solution is to exclude from L384 because that paragraph is meant only to rumen fermentation parameters.

L476-479: Make overall conclusions, but do not repeat the results.

L503: Should data be available (especially sequencing data) on a platform, such as NCBI, for instance?

Best regards.

Author Response

General points about the manuscript: The manuscript brings interesting information about the effect of mulberry silage replacing maize silage of sheep diet on growth characteristics, gastrointestinal tissue morphology, rumen fermentation parameters and bacterial diversity. This topic is relevant to comprehend the effects of inclusion levels of mulberry silage on different physiological and nutritional aspects of sheep. In my opinion the manuscript has a good aim and the manuscript is well written.

My biggest concern is that the composition of other ingredients in the diets also changed, such as maize and soybean (not only maize silage). So, in fact the study is not only “Replacement of maize silage with mulberry silage”. The other effects might confound the results and thus should be taken into account in the title and in the text. Additionally, the crude protein of the diets varied from 13.5% to 14.9%, this could be another confounding effect and authors must give a strong explanation on it.

Answer: We use mulberry silage to replace maize silage in Hu sheep diet. Since the crude protein content of mulberry silage is higher than that of maize silage, we had to reduce other feeds in an attempt to match crude progein contents. We have changed the title and text to reflect this. We do not believe that a CP change of 1% would have had a major impact on growth and physiology as both appear to have been adequate for maximum growth.

Documents similar to this article are as follows:

V Lind, Weisbjerg M R, Jrgensen G M, et al. Ruminal Fermentation, Growth Rate and Methane Production in Sheep Fed Diets Including White Clover, Soybean Meal or Porphyra sp.[J]. Animals: an Open Access Journal from MDPI, 2020, 10(1).

Buckhaus E M, Smith Z K. Effects of Corn Silage Inclusion Level and Type of Anabolic Implant on Animal Growth Performance, Apparent Total Tract Digestibility, Beef Production per Hectare, and Carcass Characteristics of Finishing Steers[J]. Animals, 2021, 11(2): 579.

Specific considerations:

Keywords: Do not use as keywords those words already mentioned in the title.

Answer: We have implemented this change.

L74: Please better describe the meaning of PM2.5, I wonder that many readers are not familiar with it.

Answer: Annotation has been added to the text.

“Mulberry trees, by contrast” is repeated twice (L75 and L79), please re-write on of the sentences in order to sound linguistically better.

Answer: We have implemented this change.

This manuscript is a bit of a mixture of British (BT) and American (AM) English, please be consistent and choose only one style. Maybe the one the Journal follows the most. Examples: Analysed (BT) vs. Analyzed (AM); fibre (BT) vs. fiber (AM); alfalfa (AM), if it would be BT spelling, then it would be lucerne.

Answer: We have adopted British English throughout.

L107: “effect of different mulberry silage on the ability of mulberry silage…” I believe that this requires proofreading and re-writing.

Answer: We have implemented this change.

L132: Please describe more clearly how the experiment was conducted and the sampling was done along the 60-day period. For example, were samples taken in the last week of this period?

Answer: We have included a better explanation.

L142: Please use “time of slaughtering”.

Answer: We have implemented this change.

L161: The factorial number is not correct formatted. Please put it as superscript. Additionally describe better what kind of inoculant you used. What are the strains in it? Homo or heterofermenter?

Answer: One gram of Chr Hansen lactococcus lactis and lactobacillus brucei powder was added to each ton of mulberry (lactic acid bacteria >1.3*1011 CFU/g)..

Table 2: Abbreviations in the table should be spelled out in footnote. Additionally, “freshweight” is written all together, please put a space.

Answer: We have implemented this change.

Fermentation quality (pH, NH3-N, VFAs and so on) of the silages were not mentioned, however this is a very important information. Is it already published? If so, please just indicate the published paper, otherwise, please include it in this manuscript.

Answer: At present, our main research purpose is to expand the application of mulberry silage in livestock production to alleviate the shortage of high-quality roughage in China and improve the utilization of unconventional roughage. Therefore, the fermentation index of the mulberry silage was not measured, but the quality of the mulberry silage used in this experiment was good from the evaluation of appearance and smell.

L193: “column” is written twice, delete one of them.

Answer: We have implemented this change.

L196: Use superscript in cm2.

Answer: We have implemented this change.

L209: Is this correct “I11nmina”? Please check and fix.

Answer: The machine model has been re-checked with the sequencing company.

L215: If this abbreviation is not used again, it’s not needed to be defined.

Answer: We have deleted the abbreviation.

L222: As there are 4 treatments, authors can also investigate linear, quadratic and cubic effects of mulberry replacement level on the diet of sheep.

Answer: The analysis of variance that we used did analyse for linear effects. As these were generally not significant, we did not anticipate any quadratic or cubic effects.

Table 3: “liveweight” is written all together, please put a space. Abbreviations in Table were defined, but not used, so better not to define them. Should SEM be spelled out in footnote? If so, please check throughout.

Answer: We have changed this..

L238: The description of treatments was already given in the Title.

Answer: We have implemented this change.

L242: VFA was already previously described. Please describe abbreviation only first time mentioned and then use just the abbreviation onward. Please check throughout for other abbreviations as well.

Answer: We have checked and implemented this change

L244: It’s missing an end period in a sentence.

Answer: We have implemented this change.

L246: This sentence “Ammonia was highest in M and H” does not match the results in Table 4, please check and fix.

Answer: We have modified the results accordingly.

Table 4: Please put the title of this table in line with the title of Table 3. The same for the titles of Tables 5 and 6.

Answer: We have implemented this change.

Table 5: Put the Duncan’s letters in superscript. Use the correct symbol for “um”. What is “V/C”? Define. Footnote is missing end period.

Answer: V/C (Villus height /Crypt depth). We have implemented this change.

Mention Figure 1 in text before inserting the Figure.

Answer: We have implemented this change.

Figure 3: Isn’t the H treatment (yellow) in the Figure? Why?

Answer: The microorganisms in group H were not significantly different from the other three groups using this analysis.

L343: “was associated”, please indicate the direction of the association (positive or negative).

Answer: We have implemented this change.

L344: “NH3-N content was positively correlated with pH and butyric acid content”, the Figure does not allow this affirmation. Please check and fix.、

Answer: We have checked this and changed the text to “the relative abundance of norank_f__F082 was positively correlated with propionic acid content (R=0.541, p<0.05) and NH3-N content (R=0.636, p<0.001). PH (R= -0.460, p <0.05) and butyric acid (R=0.487, 0.001< p <0.05) are negatively correlated with norank_f__F082.”

L357: Please check journal’s guideline on the way of including the references in the text. It’s not consistent at the moment. Please also add space before and after [ ]. Check throughout the manuscript.

Answer: We have changed the references to conform to journal requirements

L364: Please put the number 2 in CO2 subscript.

Answer: We have implemented this change.

L386: Positively? Wasn’t it negatively correlated?

Answer: The problematic part has been modified.

L391: Scientific names should be in italic. Check throughout.

Answer: We have implemented this change.

L411: Was V/C previously described?

Answer: We have implemented this change.

L465: This was already mentioned in L384. Check and fix the text in one of these places, but do not repeat it. Maybe the best solution is to exclude from L384 because that paragraph is meant only to rumen fermentation parameters.

Answer: The problematic part has been modified.

L476-479: Make overall conclusions, but do not repeat the results.

Answer: We have changed the conclusion to “The replacement of maize silage and soyabean meal with mulberry silage at the highest level improved the rumen fermentation pattern, and possibly increased the utilization rate of nitrogen”.

L503: Should data be available (especially sequencing data) on a platform, such as NCBI, for instance?

Answer: The NCBI Sequence Read Archive (SRA) database was used to deposit raw reads (Accession Number: PRJNA819287). ( Materials and Methods).

Best regards.